# Circadian rhythms modulate the effect of eccentric exercise on rat soleus muscles

Shuo-wen Chang[1,2☯], Toshinori Yoshihara[1☯], Takamasa Tsuzuki[1,3], Toshiharu Natsume[1,4], Ryo Kakigi[5], Shuichi Machida[1], Hisashi Naito[1]*

1 Graduate School of Health and Sports Science, Juntendo University, Chiba, Japan, 2 Department of Physical Education, National University of Tainan, Tainan, Taiwan, 3 Faculty of Pharmacy, Meijo University, Nagoya, Aichi, Japan, 4 School of Medicine, Tokai University, Hiratsuka, Kanagawa, Japan, 5 Faculty of Management & Information Sciences, Josai International University, Chiba, Japan

☯ These authors contributed equally to this work.
* hnaitou@juntendo.ac.jp

**Data Availability Statement:** All relevant data are within the manuscript and its Supporting Information files.

## Abstract

We investigated whether time-of-day dependent changes in the rat soleus (SOL) muscle size, after eccentric exercises, operate via the mechanistic target of rapamycin (mTOR) signaling pathway. For our first experiment, we assigned 9-week-old male Wistar rats randomly into four groups: light phase (zeitgeber time; ZT6) non-trained control, dark phase (ZT18) non-trained control, light phase-trained, and dark phase-trained. Trained animals performed 90 min of downhill running once every 3 d for 8 weeks. The second experiment involved dividing 9-week-old male Wistar rats to control and exercise groups. The latter were subjected to 15 min of downhill running at ZT6 and ZT18. The absolute (+12.8%) and relative (+9.4%) SOL muscle weights were higher in the light phase-trained group. p70S6K phosphorylation ratio was 42.6% higher in the SOL muscle of rats that had exercised only in light (non-trained ZT6). Collectively, the degree of muscle hypertrophy in SOL is time-of-day dependent, perhaps via the mTOR/p70S6K signaling.

## Introduction

Light-responsive central clock genes in the suprachiasmatic nucleus (SCN) coordinate circadian rhythms through signaling of peripheral clock genes in other tissues [1]. Glucocorticoids are the primary humoral mediators that send the reset signal from the SCN to the peripheral clock [2]. Hence, organ function varies depending on the time of day. As the main contractile tissue of the body involved in movement, skeletal muscle is not only important for posture but also plays a major role in energy metabolism and glucose uptake [3]. In our previous study, we demonstrated circadian variation in intracellular protein synthesis signaling in rat skeletal muscles [4], similar to what has been observed in body temperature, locomotor activity, and hormone secretion [5, 6]. Specifically, we observed diurnal variation in mechanistic (mammalian) target of rapamycin (mTOR)/70 kDa ribosomal protein S6 kinase 1 (p70S6K) signaling and extracellular signal-regulated kinase (ERK) signaling in skeletal and cardiac muscles. These pathways are key mechanisms for skeletal muscle hypertrophy under mechanical

**Funding:** This work was supported by the Japan Society for the Promotion of Science KAKENHI under Grant number 17K13192 to S.C. and Grant number 17K01765 to T.Y.; and Ministry of Education, Culture, Sports, Science and Technology Supported Program for the Strategic Research Foundation at Private Universities. The funders had no role in study design, data collection and analysis, decision to publish, or preparation of the manuscript.

**Competing interests:** The authors have declared that no competing interests exist.

overload (e.g., strength training). Thus, mTOR/p70S6K and ERK signaling pathways are potential targets for maintenance or increase of skeletal muscle mass (muscle hypertrophy). This therapeutic strategy has numerous applications, from improving the health of the elderly to boosting athlete performance.

Given the influence of circadian rhythms, treatment efficacy and the probability of adverse drug reactions will most likely depend on the time of day [7, 8]. Thus, there may be an optimal time for activating mTOR signaling to maximize the effects of exercise on muscle tissue. Recent studies on human skeletal muscle have investigated how exercise at different times of day influences muscle mass (muscle fiber size), with inconsistent results. For instance, following 24 weeks of combined strength and endurance training, evening groups (16:30–18:30) experienced a greater gain in muscle mass than morning groups (07:30–09:30) [9]. In contrast, muscle mass gain did not differ between morning (07:00–09:00) and evening (17:00–19:00) groups after 10 weeks of strength training [10]. Multiple factors could generate such inconsistency, including variation in individual physical fitness, training programs, daily timing of exercises, and diet. Notably, these studies did not describe their criteria for determining when training should occur nor did they consider subject diet during training periods. Therefore, in this study, we designed a rat experiment that controlled the effects of important confounding factors (e.g., individual variation, protein intake, exercise intensity, and exercise timing). Our aims were to examine 1) the effect of exercise on muscle hypertrophy at different times of day and 2) the underlying mechanisms in rat soleus (SOL) muscle. We hypothesized that eccentric exercise at different periods in the day (light vs. dark) will influence mTOR signaling activation and subsequent training-induced skeletal muscle hypertrophy.

## Materials and methods

### Experimental animals

Nine-week-old male Wistar rats were obtained from a licensed laboratory animal vendor (SLC Inc., Hamamatsu, Shizuoka, Japan). Water and food were provided *ad libitum*. All rats were housed in an environmentally controlled room (temperature: 23 ± 1˚C; relative humidity: 55% ± 5%; 12/12 h light/dark cycle, with lights on at 18:00 and off at 6:00) after acclimation for 1 week, prior to the experiment. All procedures were approved by the Juntendo University Animal Care and Use Committee (H29-07).

### Experiment 1: Effect of eccentric exercise on muscle hypertrophy at different times of the day for 8 weeks

Twenty-four male Wistar rats were randomly assigned to two groups: Zeitgeber Time (ZT) 6 (light phase; L, n = 12) and ZT18 (dark phase; D, n = 12) (ZT0: lights on at 18:00, ZT12: lights off at 6:00). Each group was further divided into untrained control (C) or trained (TR) groups (n = 6 per subgroup). The training program was based on previously published methods [11]. Once every 3 d (20 sessions in total), TR animals participated in 90 min of downhill running on an animal treadmill (incline: -16˚, speed: 16 m/min). Forty-eight hours (including 12 h of fasting) after the last training session, rats were anesthetized with isoflurane (3%–5%) and pentobarbital sodium (50 mg/kg). Right and left SOL muscles and epididymal white adipose tissue was immediately removed and weighed. Thereafter, muscle samples were gently stretched to near optimal length using a compass before being frozen in liquid nitrogen and stored at -80˚C.

## Experiment 2: Effect of eccentric exercise on mTOR signaling in SOL at different times of the day

Forty-two male Wistar rats were randomly assigned to two groups: ZT6 (L, n = 21) and ZT18 (D, n = 21). These periods were selected because our previous study showed that mTOR/p70S6K signaling in skeletal muscles is high at ZT6 and low at ZT18 [4]. Rats from each group were further divided into before (Pre), immediately after (Pt0), and 1 h after eccentric exercise (Pt1) (n = 7 per category).

Following 12 h of fasting, Pt0 and Pt1 rats participated in a bout of downhill running for 15 min on an animal treadmill (incline: -16˚, speed: 16 m/min). At appropriate time points (Pre, Pt0, Pt1), rats were anesthetized with isoflurane (3%–5%) and pentobarbital sodium (50 mg/kg). Muscle samples were frozen immediately in liquid nitrogen and stored at -80˚C until analysis.

Blood samples were collected from the inferior vena cava, centrifuged at $1300 \times g$ for 10 min to obtain serum and stored at -80˚C. Corticosterone concentrations were estimated by commercial laboratories (Shibayagi Co., Ltd., Gunma, Japan and Oriental Yeast Co., Ltd., Tokyo, Japan).

## Measurement of SOL myofiber cross-sectional area (CSA) and centrally nucleated myofibers

Muscle section preparation and staining were performed following previously described techniques [12]. Briefly, frozen SOL samples were sliced into 10 μm sections using a cryostat (CM3050S, Leica, Wetzlar, Germany) at -20˚C, and subsequently stained with hematoxylin and eosin (H&E). Section images were captured using a microscope (10×; BZ-8000; Keyence, Osaka, Japan) and transferred to a computer. CSA of 200–250 muscle fibers were randomly measured using the ImageJ software (NIH, Bethesda, MD, USA). The central nucleus was used as an indicator of muscle regeneration. We evaluated the ratio of the centronucleated fiber, defined by the centrally nucleated muscle fibers divided by the total number of muscle fibers.

## Western blotting analysis and immunodetection

SOL samples were frozen in liquid nitrogen, powdered, and then homogenized in ice-cold buffer (50 mM HEPES [pH 7.4], 1 mM EDTA, 1 mM EGTA, 20 mM β-glycerophosphate, 1 mM $Na_3VO_4$, 10 mM NaF, and 1% Triton X-100) containing a protease inhibitor cocktail (cOmplete EDTA-free; Roche, Penzberg, Germany) and phosphatase inhibitor cocktail (PhosSTOP; Roche). Homogenates were centrifuged at $12000 \times g$ at 4˚C for 15 min. Supernatant protein concentrations were determined using a BCA Protein Assay Kit (Thermo, Rockford, IL, USA). Protein extracts were solubilized in sample buffer (30% glycerol, 5% 2-mercaptoethanol, 2.3% SDS, 62.5 mM Tris–HCl [pH 6.8], and bromophenol blue) and incubated at 95˚C for 5 min. Samples containing total protein were subsequently loaded onto 10–12% TGX Fast Cast acrylamide gels (Bio-Rad Laboratories, Hercules, CA, USA) and electrophoresed at 150 V for 45 min. Separated proteins were transferred to PVDF membranes (Bio-Rad Laboratories) using a Bio-Rad Mini Trans-Blot cell at 100 V and 4˚C for 60 min in transfer buffer (25 mM Tris, 192 mM glycine, and 20% methanol).

Thereafter, membranes were blocked for 1 h using blocking buffer (5% nonfat dry milk in Tween-Tris-buffered saline [T-TBS: 40 mM Tris-HCl, 300 mM NaCl, and 0.1% Tween 20, pH 7.5]) at room temperature (25–26˚C). Membranes were subsequently incubated for 2 h at room temperature (25–26˚C) with the following primary antibodies (all from Cell Signaling Technology, Danvers, MA, USA): anti-phospho-mTOR Ser2448 (1:2000; #2971), anti-mTOR

(1:2000; #2972), anti-phospho-p70S6K Thr389 (1:2000; #9234), anti-p70S6K (1:2000; #9202), anti-phospho-4EBP1 Thr37/46 (1:2000; #9459), anti-4EBP1 (1:2000; #9644), anti-phospho-p44/42 ERK1/2 Thr202/Thr204 (1:5000; #4370), and anti-p44/42 ERK1/2 (1:5000; #4695) in Can Get Signal, a dilution buffer (Toyobo). Thereafter, membranes were incubated with secondary antibodies (1:5000; #7074) in Can Get Signal for 1 h at room temperature (25–26˚C). Signals were detected using Immobilon Western Chemiluminescent HRP Substrate (Millipore Corporation) and recorded with a ChemiDoc™ Touch imaging system (Bio-Rad). Signal intensity was analyzed using Image Lab v.5.2.1 (Bio-Rad). The ratio of phosphorylated to total protein expression was determined using arbitrary units.

## Statistical analysis

All values are presented as the mean ± standard error (SE). Absolute muscle weight strongly correlated with body weight; therefore, relative skeletal muscle weight was calculated by dividing absolute muscle weight by body weight. Group differences were analyzed using two-way analysis of variance (ANOVA), one-way analysis of variance (ANOVA), or unpaired t-test. When an interaction (time of day × exercise) was observed, Bonferroni's post-hoc test was performed. Significance was set at $P < 0.05$. All analyses were performed using GraphPad Prism version 6.0 (GraphPad Software Inc., La Jolla, CA, USA).

## Results

### Experiment 1: Effect of eccentric exercise on muscle hypertrophy at different times of the day for 8 weeks

Time of day × exercise interactions were absent for body weight ($P = 0.6434$; **Fig 1A**), fat weight ($P = 0.7170$; **Fig 1B**), and average food intake ($P = 0.5487$; **Fig 1C**). The fat weight in rats in the L(D)TR group was significantly decreased compared with that of rats in the L(D) control (C) group (**Fig 1B**). No difference was observed in body weight (**Fig 1A**) and average food intake between the groups (**Fig 1C**). The weekly food intake in the 1st and 2nd weeks was significantly lower in L(D)TR rats than in L(D)C rats (**Fig 1D**).

We assessed time of day × exercise interaction effects on the plantaris and gastrocnemius muscle weights (**Table 1**). No such effects could be observed. Time of day × exercise interaction was significant for absolute ($P < 0.05$; **Fig 2A**) and relative ($P < 0.05$; **Fig 2B**) SOL weights, as well as for SOL fiber CSA ($P < 0.05$; **Fig 2C**). Control groups did not differ in absolute and relative SOL weights at different times of the day. However, between the trained animals, LTR rats had significantly heavier muscles than DTR rats (absolute muscle weight, +12.8%, $P < 0.001$; relative muscle weight, +9.4%, $P < 0.05$; **Fig 2A and 2B**). We observed no significant difference in CSA between LTR and DTR rats, although the former was slightly larger (+5.3%, $P = 0.4748$; **Fig 2C**). Muscle fiber CSA was significantly greater in LTR rats than in LC rats (+17.9%, $P < 0.0001$), but that in DTR and DC rats did not differ ($P = 0.5221$), as shown in **Fig 2C**. The representative histological findings of the SOL muscles from each group are shown in **Fig 2D**. No time of day × exercise interaction effects on the prevalence of centronucleated fibers in soleus muscle could be observed ($P = 0.9080$; **Fig 2E**).

### Experiment 2: Effect of eccentric exercise on mTOR signaling in SOL at different times of the day

Time of day and exercise significantly affected ($P < 0.05$ and $P < 0.01$, respectively) mTOR phosphorylation, which was higher in LPt0 rats than in LPre rats after acute eccentric exercise (**Fig 3A**). Although we did not observe time of day × exercise interaction effects on the

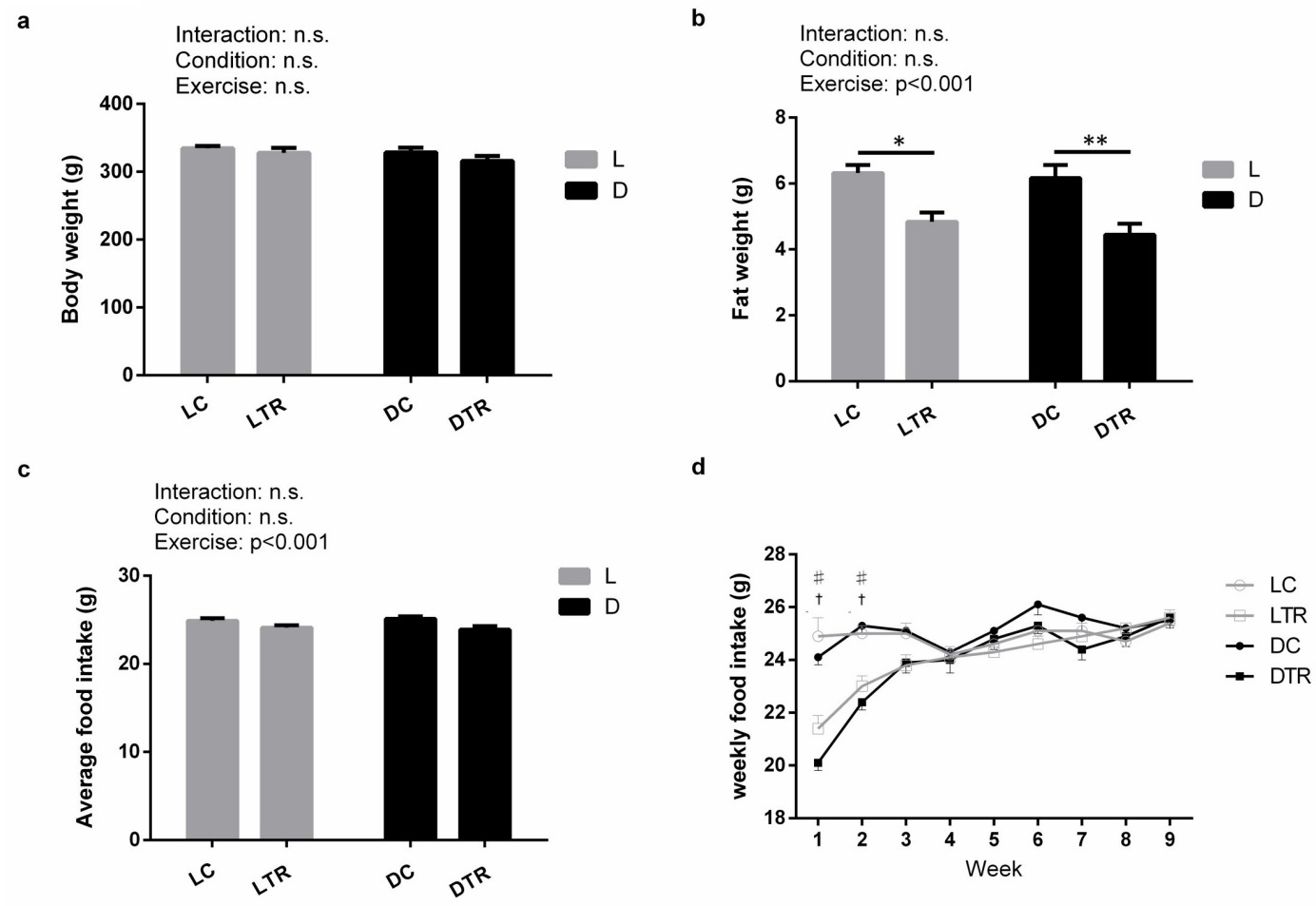

**Fig 1. Body weight, fat weight, and food intake.** Body weight (a), fat weight (epididymal white adipose tissue) (b), average food intake (c), and weekly food intake (each value represents the mean of 6 days) (d). L, light phase; D, dark phase; C, control; TR, trained group. Values are the mean ± standard error (SE); n = 6 per group. The results of the two-way analysis of variance (ANOVA) are displayed in a, b, c; *$P < 0.05$, **$P < 0.01$. The results of the one-way analysis of variance (ANOVA) are displayed in d; [†]LC versus LTR, $P < 0.05$; [♯]DC versus DTR, $P < 0.05$.

phosphorylation ratios of mTOR ($P = 0.4291$; **Fig 3A**), 4EBP1 ($P = 0.7991$; **Fig 3C**), and ERK1/2 ($P = 0.9119$; **Fig 3D**), p70S6K phosphorylation significantly changed after exercise ($P < 0.001$; **Fig 3B**) and was higher in LPt0 rats than in DPt0 rats (+42.6%, $P < 0.01$; **Fig 3B**).

**Table 1. Effect of eccentric exercise on plantaris and gastrocnemius muscle weight at different times of the day for 8 weeks.**

|  | L | | D | |
|---|---|---|---|---|
|  | C | TR | C | TR |
| Plantaris muscle weight, mg | 330.7 (17.3) | 343.7 (15.5) | 324.5 (13.4) | 331.4 (24.0) |
| Plantaris muscle weight, mg g BW$^{-1}$ § | 0.99 (0.03) | 1.05 (0.03) | 0.99 (0.02) | 1.05 (0.03) |
| Gastrocnemius muscle weight, mg | 1612.8 (44.2) | 1643.4 (73.8) | 1536.6 (75.7) | 1595.8 (100.2) |
| Gastrocnemius muscle weight, mg g BW$^{-1}$ § | 4.8 (0.1) | 5.0 (0.1) | 4.8 (0.1) | 5.1 (0.1) |

Values are means (standard error; SE). Two-way ANOVA demonstrated that there was no time of day × exercise interaction. n = 6 per group. § represents a main effect of training. L, light phase; D, dark phase; C, control; TR, trained group.

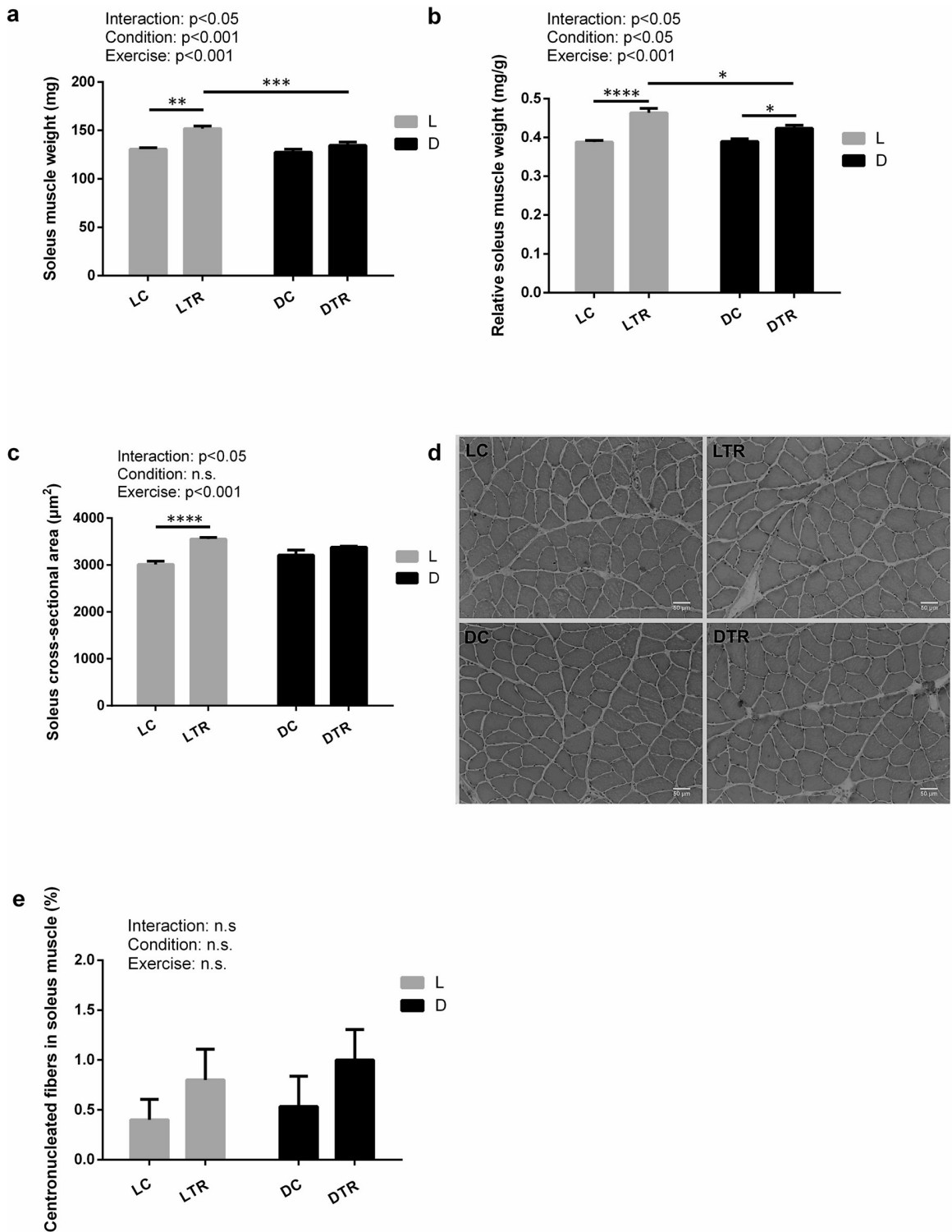

**Fig 2. Effect of eccentric exercise on soleus muscle at different times of the day for 8 weeks.** Soleus muscle weight (a), soleus muscle weight relative to body weight (b), soleus cross-sectional area (CSA) (c), hematoxylin and eosin (H&E) staining of rat soleus muscle sections (10× magnification, scale bar = 50 μm) (d), centronucleated fibers of the soleus muscle (e). L, light phase; D, dark phase; C, control; TR, trained group. Values are the mean ± standard error (SE); n = 6 per group. The results of the two-way analysis of variance (ANOVA) are displayed. $^{*}P < 0.05$, $^{**}P < 0.01$, $^{***}P < 0.001$, $^{****}P < 0.0001$.

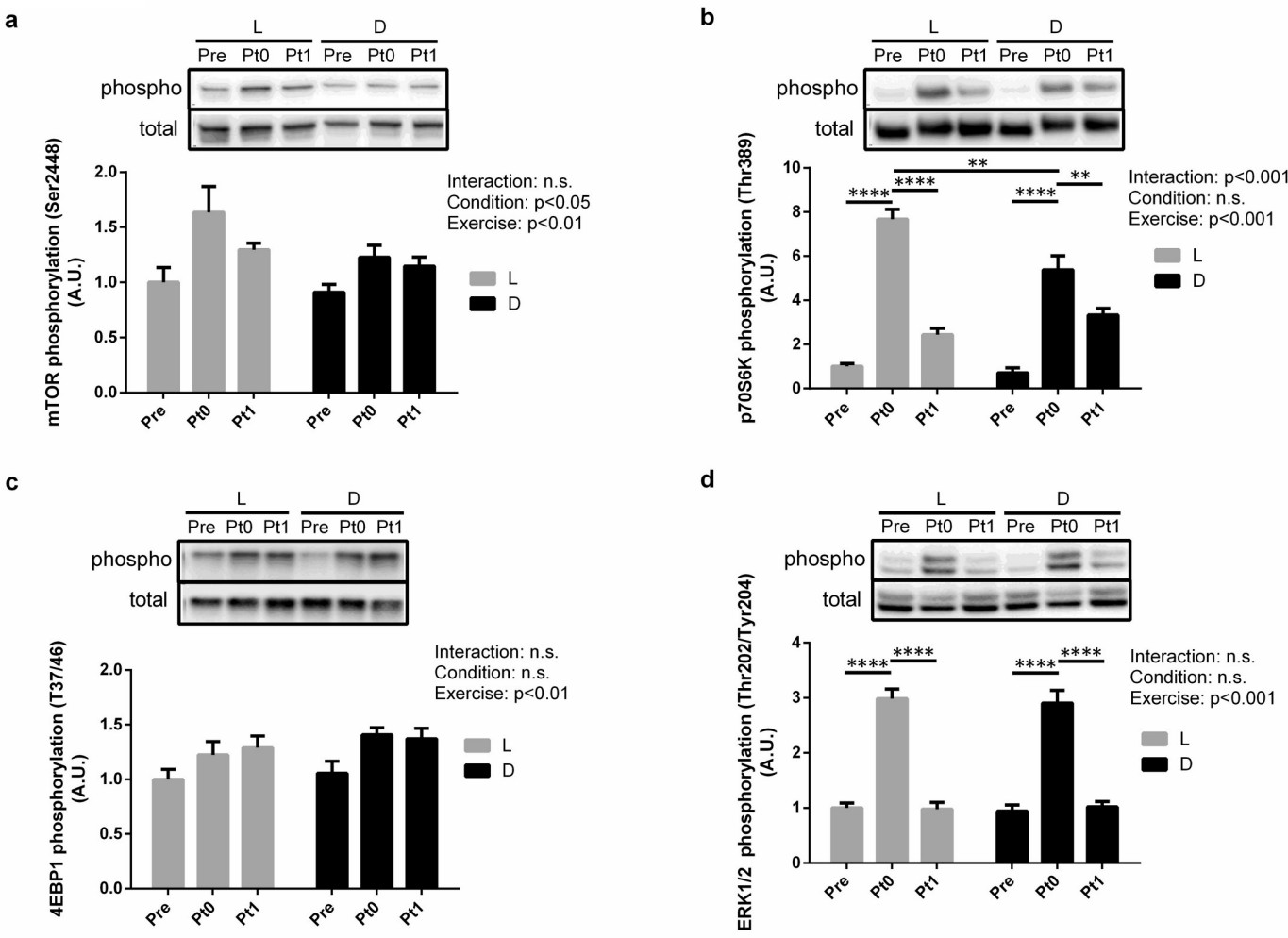

**Fig 3. Effect of eccentric exercise on mTOR signaling in soleus muscle at different times of the day.** Phosphorylation rates of mTOR (a), p70S6K (b), 4EBP1(c), and ERK1/2 (d) in the soleus muscle. Samples were collected before (Pre), immediately after (Pt0), and 1 h after (Pt1) eccentric exercise. L, light phase; D, dark phase. Values are the mean ± standard error (SE); n = 7 per group. The results of the two-way analysis of variance (ANOVA) are displayed. $^{**}P < 0.01$, $^{****}P < 0.0001$.

Exercise significantly affected ($P < 0.001$) ERK1/2 phosphorylation, which was higher in L(D) Pt0 rats than in L(D)Pre and L(D)Pt1 rats after acute eccentric exercise (**Fig 3D**).

Time of day and exercise significantly affected ($P < 0.001$ and $P < 0.0001$, respectively) serum corticosterone concentration, but we did not observe time of day × exercise interaction effects on the serum corticosterone concentration ($P = 0.4379$; **Fig 4A**). Exercise significantly affected serum corticosterone concentration, which was higher in L(D)Pt0 rats than in L(D) Pre rats and higher in DPt0 rats than in DPt1 (**Fig 4A**). We observed no significant difference in serum corticosterone concentration between LPt0 and DPt0 rats, but the increased rate of serum corticosterone immediately after eccentric exercise in the light phase group ($P < 0.05$; **Fig 4B**) was significantly lower than that in the dark phase group.

## Discussion

The main finding of our study was that 8 weeks of eccentric training in the light phase induced a greater increase in SOL mass and size than in the dark phase. This difference may be

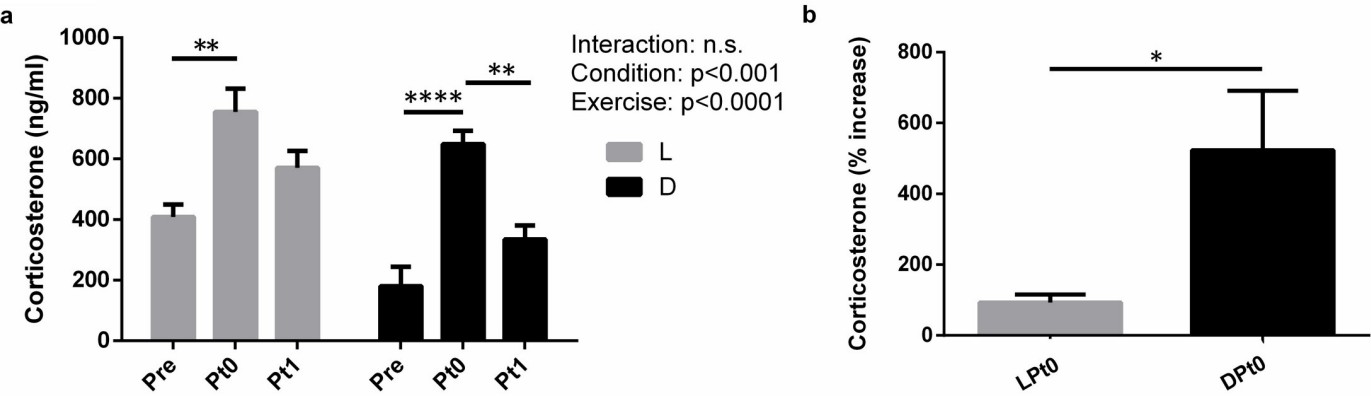

**Fig 4. Increased rate of serum corticosterone immediately after eccentric exercise.** Serum corticosterone concentration (a), rate of increase in serum corticosterone (b). L, light phase; D, dark phase. Values are the mean ± standard error (SE); n = 7 per group. The results of the two-way analysis of variance (ANOVA) are displayed in a. The results of the unpaired t-test are displayed in b. $^*P < 0.05$, $^{**}P < 0.01$, $^{****}P < 0.0001$.

associated with greater phosphorylation of proteins involved in mTOR/p70S6K signaling. These data support previous research in humans, demonstrating that after 24 weeks of combined strength and endurance training, exercising at different times of day may affect the degree of muscle hypertrophy [9].

In this study, using the same low-intensity downhill running protocol as in the previous study, the same results were seen, with no difference in body weight after training in each group [11]. A significant increase in body fat was observed in the control group but not in the exercise group, confirming the effect of exercise training on suppressing the increase in fat mass. This result is the same as that in the previous study (low-intensity downhill running; incline less than -16˚, speed lower than 16 m/min) [13]. During the first two weeks of this study, the average weekly intake in the training group was lower than that in the control group. As was also observed under this situation in previous studies, the intake in the low-intensity exercise group, compared to that in the no-exercise group, first dropped and then returned to the intake in the no-exercise group [14]. It is thought that exercise-induced appetite suppression is due to increased catecholamines associated with exercise stress [15] or increased release of corticotropin-releasing factors (appetite-suppressing peptide) by the hypothalamus [16].

Although acute signaling response does not always yield chronic effects, previous studies on rat skeletal muscle have demonstrated that the activation of p70S6K signaling is associated with hypertrophic effects of chronic exercise training [17, 18]. Therefore, the fact that mTOR/p70S6K signaling was higher in the light phase than in the dark phase suggests that circadian rhythms are part of the mechanisms underlying exercise-related increase in muscle mass and size. Notably, however, in a previous human study, it was found that time of the day did not influence resistance-exercise-induced p70S6K phosphorylation [19]. Therefore, further research is required to clarify precise mechanisms of exercise-related skeletal muscle hypertrophy, which might explain the discrepancies across studies.

ERK signaling has been suggested to play a role in the signaling network required for regeneration and hypertrophy [20] in the skeletal muscle. Furthermore, ERK signaling pathway is activated by exercise, depending on its intensity [21]. Previous research demonstrated that resistance exercise caused an increase in phosphorylation of the ERK signaling in rat and human skeletal muscle [22, 23]. In this study, ERK signaling was also increased after acute exercise, but no difference was observed at different periods in the day (light vs. dark). This

result indicates that ERK signaling is an unlikely pathway for exercise-induced muscle hypertrophy at different periods. Nevertheless, the results of ERK signaling in this study can prove that the intensity of exercise performed at different periods is the same.

In previous studies, it was observed that 4EBP1 signaling after resistance exercise was unchanged compared to that in the case of no exercise [24, 25]. However, recent studies have shown that after resistance exercise, 4EBP1 signaling was significantly higher than the signaling without exercise [26, 27]. Therefore, the effect of exercise on 4EBP1 is still unknown. In the present study, there was no change in 4EPB1 with exercise at different times. This result also indicates that 4EBP1 signaling is unlikely to be the pathway for exercise-induced muscle hypertrophy at different times. In contrast, the result of p70S6K signaling is different in this study; differential activation of p70S6K signaling after eccentric exercise was key in the SOL muscle hypertrophy.

Little is known about the differential p70S6K activation at different periods in the day; however, this may be due to the effect of serum corticosterone. Serum corticosterone may be associated with greater activation of mTOR/p70S6K in the light phase than in the dark phase. Interestingly, in a previous study, it was demonstrated that glucocorticoid (corticosterone in rats, cortisol in humans) levels regulate p70S6K phosphorylation, with higher concentrations suppressing phosphorylation in L6 skeletal myoblasts [28]. There are two possible mechanisms to explain the effect of corticosterone on the mTOR signaling pathway. The first is that glucocorticoids activate glucocorticoid receptors (GR), leading to a decrease in the concentration of intracellular branched-chain amino acids (BCAAs) via increased expression of branched-chain amino transferase 2 (BCAT2) by KLF15, and that via Ras homolog enriched in brain (Rheb) leads to a decrease in the mTOR activity [29, 30]. In the second pathway, glucocorticoids promote the assembly of the TSC1-2 (hamartin-tuberin) complex by regulating the development and DNA damage responses 1 (REDD1) through activation of the GR, and finally reduce the mTOR activity via Rheb [30–32].

The effect of corticosterone in rodents is likely linked to circadian rhythm, as muscle protein breakdown occurs 4 h after the plasma corticosterone concentration peaks [33] at approximately midday (between ZT11 and ZT13, or at the end of the inactive phase and beginning of the active phase) [4, 34]. In this study, the serum corticosterone concentration in the light period (ZT6) group was generally higher than that in the dark period (ZT18) group, possibly due to the interconnection between circadian rhythm and corticosterone concentration that peaked around ZT11-13. In contrast, the corticosterone peak in the dark period (ZT18) group was past and the concentration had lowered. The serum corticosterone levels in the light phase and dark phase groups immediately after exercise were not significantly different, but the rate of increase in corticosterone concentration immediately after exercise was greater in the dark phase than in the light phase group. These observations tentatively suggest that corticosterone has a slightly antagonistic impact on mTOR/p70S6K signaling during the light phase, explaining the increase in exercise-induced p70S6K phosphorylation and subsequent muscle hypertrophy during that time of the day.

Increased circadian dependence of serum corticosterone may be associated with a 1 h exposure to light in the dark phase [35]. This may reduce the effectiveness of exercise in the dark phase under light exposure. However, to block the light in this study, we covered the treadmill using a black shielding bag for 8 weeks during the exercise experiment for the DTR group, and the room temperature was monitored. Therefore, it is believed that the increase in corticosterone due to light exposure at night can be suppressed.

Previous studies have shown a relationship between mTOR/p70S6K and muscle protein synthesis. A mouse study by Ogasawara et al. [36] reported that the promotion of muscle protein synthesis by resistance exercise (transcutaneous electrical stimulation to induce muscle

contraction) is regulated by mTOR (both rapamycin-sensitive mTORC1 and rapamycin-insensitive mTORC1 or mTORC2). However, a recent study reported that exercise-induced protein synthesis is independent of mTOR/p70S6K. You et al. [37] reported that protein synthesis by mechanical loading is mTORC1-independent in the case of myotenectomy-induced muscle hypertrophy in mice. Different experimental methods of inducing muscle hypertrophy-like electrical stimulation or myotenectomy-induced muscle hypertrophy may also result in different pathways of muscle protein synthesis. Therefore, the relationship between mTOR/p70S6K and muscle protein synthesis remains unclear. In this study also, we have not elucidated the relationship between mTOR/p70S6K as we have not measured muscle protein synthesis.

Our current findings contradict the results of a previous study demonstrating that 6 weeks of aerobic training (swimming for 60 min/bout, 5 bouts/week) increased relative gastrocnemius muscle weight, more so in the dark phase than in the light phase [38]. Differences in exercise type and muscle fiber type between studies could explain the discrepancy. Previous research indicated that different exercises irrefutably activate specific intracellular signaling pathways [39, 40]. Moreover, hormones as indicators of systemic factors also affect muscle mass; for example, differences in hormone levels in the body (testosterone:corticosterone ratio) or differences in hormone levels caused by exercise. Therefore, the effect of systemic factors on muscle mass cannot be ignored [41, 42].

Although previous studies in human subjects have demonstrated that the timing of feeding post-exercise (1–3 hours after training) plays an important role in promoting protein synthesis [43], it has recently been shown that training increases the sensitivity of muscle protein synthesis for at least 24 hours [44, 45]. The intensity and volume of exercise also influence the increase in post-exercise feeding-related protein synthesis [44, 46]. However, previous studies have shown that the effect of post-exercise feeding timing on muscle protein synthesis is attenuated when exercise does not take place until exhaustion [44]. In this study, we used low-intensity exercise [11, 47]; therefore, we believe that increase in muscle mass due to post-exercise feeding might be difficult.

Although previous research found differences in food intake between trained groups at different periods in the day (light vs. dark) [38], the daily food consumption of the trained groups used in this study at different periods in the day did not differ despite the rats having *ad libitum* access to water and food. Therefore, unlike in the previous study, we can better associate the muscle weight with exercise, rather than with changes in body weight due to differential food intake. As exercise type (endurance or resistance) affects muscles differently, future studies should consider this parameter before investigating whether circadian rhythms definitively affect the exercise-induced elevation of mTOR/p70S6K signaling and associated muscle hypertrophy.

The previous study on rats has shown that most of the fiber components of the soleus muscle are mobilized during downhill running (incline: -16˚) [48]. In addition, when low-intensity exercise (10–20 m/min) is performed with downhill running (treadmill grade 15% decline), fast-twitch muscles might be involved to a much lesser extent in the motor unit recruitment [49]. Therefore, the cause of the induced muscle hypertrophy only in the soleus muscle could be determined as only the low-intensity exercise was involved in this study. It is thought that low-intensity exercise may not reflect the effects of circadian rhythm on fast-twitch muscles as the fast-twitch muscles are scarcely stimulated low-intensity exercise. In the future, employment of high-intensity exercise will be required to prove this.

Upon the onset of muscle damage, satellite cells are activated; they proliferate and differentiate, eventually fusing with other satellite cells to form myofibers with central nuclei [50, 51]. Therefore, the central nuclei (shown as the ratio of the centronucleated fiber) are used as an

indicator of regenerating muscle. It was previously reported that the central nucleus of the soleus muscle of rats was clearly observed after 5 days of a bout of downhill running [51]. However, in this study, there was no difference in the central nucleus in any of the groups because we used the same training intensity and training time for 8 weeks. Therefore, the muscles would be accustomed to the training stimulus, and damage would be less likely to occur.

The clock gene has been reported to be involved in the structure and function of skeletal muscle [52]. It has also been demonstrated that exercise affects the circadian clock (Bmal1/CLOCK) of skeletal muscle [53]. Based on these studies, it may be possible to influence muscles from exercise via the clock genes. As per recent studies, some pathways are likely to influence Per1 by glucocorticoids via REDD1 [54]. Another pathway possibly affects BMAL1 through the activation of mTORC1 [55]. However, further research is needed to confirm whether these pathways affect muscle mass.

We acknowledge that we did not assess changes in mTOR/p70S6K signaling during exercise periods, as we focused only on time points immediately after one exercise session, and at the end of an 8-week session. Future work, including time points during the exercise period, should provide a deeper understanding of the mTOR/p70S6K involvement. We also did not directly measure muscle protein synthesis rate after exercise. Thus, we cannot be certain that enhancement in mTOR/p70S6K signaling directly led to post-exercise muscle hypertrophy. Despite these limitations, our data provide novel insights into the potential influence of circadian rhythms when determining the effectiveness of eccentric exercise in the gaining of muscle mass and size. Future studies are required to clarify the effects of corticosterone administration on protein synthesis (mTOR/p70S6K signaling) following eccentric exercise.

## Conclusions

In conclusion, we suggest that eccentric exercise training at different times of the day affects the extent of muscle hypertrophy in rat SOL, a change potentially mediated by differential phosphorylation of the mTOR/p70S6K signaling pathway. The circadian rhythm of corticosterone is a prime candidate for the regulation of differential p70S6K activation in response to eccentric exercise throughout the day.

## Supporting information

**S1 Raw images.**
(PDF)

**S1 File.**
(XLSX)

## Author Contributions

**Conceptualization:** Shuo-wen Chang, Toshinori Yoshihara, Hisashi Naito.

**Data curation:** Shuo-wen Chang, Toshinori Yoshihara, Hisashi Naito.

**Formal analysis:** Shuo-wen Chang, Toshinori Yoshihara.

**Funding acquisition:** Shuo-wen Chang.

**Investigation:** Shuo-wen Chang, Toshinori Yoshihara, Takamasa Tsuzuki, Toshiharu Natsume.

**Methodology:** Shuo-wen Chang, Toshinori Yoshihara, Hisashi Naito.

**Project administration:** Shuo-wen Chang.

**Supervision:** Hisashi Naito.

**Visualization:** Shuo-wen Chang.

**Writing – original draft:** Shuo-wen Chang, Toshinori Yoshihara, Takamasa Tsuzuki, Toshiharu Natsume, Ryo Kakigi, Shuichi Machida, Hisashi Naito.

**Writing – review & editing:** Shuo-wen Chang, Toshinori Yoshihara, Takamasa Tsuzuki, Toshiharu Natsume, Ryo Kakigi, Shuichi Machida, Hisashi Naito.

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
