## [Decision Letter · Decision Letter 0]

11 Jun 2021

PONE-D-21-07409

Circadian rhythms modulate the effect of eccentric exercise on rat skeletal muscles via the mTOR signaling pathway

PLOS ONE

Dear Dr. Naito,

Thank you for submitting your manuscript to PLOS ONE. After careful consideration, we feel that it has merit but does not fully meet PLOS ONE’s publication criteria as it currently stands. Therefore, we invite you to submit a revised version of the manuscript that addresses the points raised during the review process.

We look forward to receiving your revised manuscript.

Kind regards,

Atsushi Asakura, Ph.D

Academic Editor

PLOS ONE

Journal Requirements:

2)  PLOS ONE now requires that authors provide the original uncropped and unadjusted images underlying all blot or gel results reported in a submission’s figures or Supporting Information files. This policy and the journal’s other requirements for blot/gel reporting and figure preparation are described in detail at https://journals.plos.org/plosone/s/figures#loc-blot-and-gel-reporting-requirements and https://journals.plos.org/plosone/s/figures#loc-preparing-figures-from-image-files. When you submit your revised manuscript, please ensure that your figures adhere fully to these guidelines and provide the original underlying images for all blot or gel data reported in your submission. See the following link for instructions on providing the original image data: https://journals.plos.org/plosone/s/figures#loc-original-images-for-blots-and-gels.

Reviewers' comments:

Reviewer's Responses to Questions

**Comments to the Author**

1. Is the manuscript technically sound, and do the data support the conclusions?

Reviewer #1: Partly

Reviewer #2: Yes

2. Has the statistical analysis been performed appropriately and rigorously? 

Reviewer #1: Yes

Reviewer #2: Yes

3. Have the authors made all data underlying the findings in their manuscript fully available?

Reviewer #1: Yes

Reviewer #2: Yes

4. Is the manuscript presented in an intelligible fashion and written in standard English?

Reviewer #1: Yes

Reviewer #2: Yes

5. Review Comments to the Author

Reviewer #1: This study aimed to investigate the effect of circadian rhythms on exercise-mediated muscle anabolism. The authors concluded that the degree of eccentric exercise-mediated muscle hypertrophy in soleus muscle is time-of-day dependent and circadian rhythms of corticosterone and mTOR/p70S6K signaling affect this adaptation. Although this is an interesting project that will bring important insight to our field, there are many concerns.

L46-48: The authors do not follow the current dogma that mTORC1/p70S6K is not a major signaling pathway in the regulation of contraction-induced protein synthesis. Also, ref. 10 does not report the correlation between signaling and muscle hypertrophy. The authors should revise this point and further add the reference that ERK signaling contributes to the contraction-induced protein synthesis.

L68: Why did the authors only investigate soleus muscle? This point significantly reduces the value of this paper. Generally, downhill running affects not only soleus muscle but also gastrocnemius (and plantalis) muscle and therefore the authors should add the data of gastrocnemius at least the results of chronic training.

L84-87: Was exercise intensity constant?

L95-96: Ref. 4 (your previous study) reported that mTOR signaling is high at ZT6 as compared with at ZT18 in plantaris but not soleus.

L100: Why did the authors use different exercise protocols between acute and chronic exercise?

L123-: Is 99% glycerol correct? What does “samples containing total protein” mean?

L156-162: Why is there no difference in body weight between the control and exercise training groups when the control rats eat a lot and do not exercise? Need discussion and add fat mass data if possible.

L187-191: mTOR S2448 phosphorylation does not necessarily associate mTORC1 activity. The authors can evaluate mTORC1 activity more appropriately to measure both p70S6K and 4E-BP1 phosphorylation.

L202-204: The authors report only a %change in serum corticosterone concentration. Please add absolute values at pre- and post-exercise.

L217-226: Resistance exercise increases mTORC1 signaling for more than 24h after exercise while endurance exercise activates mTORC1 signaling only immediately after exercise. The results of this study are in accordance with the signaling responses of endurance exercise rather than resistance exercise, suggesting that factors other than mTORC1 signaling associate downhill running-induced muscle hypertrophy.

L228: Ref. 4 does not report the role of ERK signaling in the mechanisms of muscle hypertrophy.

L236-253: Regarding corticosterone, the authors state that serum corticosterone may be associated mTOR/p70S6K signaling, as it has a role in muscle protein breakdown. Please discuss more specifically with the references. How muscle protein breakdown affects mTOR signaling?

Serum corticosterone response may affect the mTOR signaling response but how about the resting condition? Not resting (absolute value) but only responses affect mTOR signaling?

It is interesting and important to measure REDD1 to know the mechanisms of reduced mTORC1 responses.

L260-265: Obviously, exercise type and fiber type affect the signaling responses. However, the authors suggested that different adaptation to the training is led by serum corticosterone (systemic factor). Therefore, the authors should discuss not only signaling but also systemic factors.

L268: Which paper shows that the timing of feeding is important to induce muscle hypertrophy?

L269: Ref. 30 is a review paper. So please add the original article(s).

Reviewer #2: Please if it is possible to show the circadian clock proteins or the gene expression, since exercise could have modified your circadian clock (i.e Bmal1 / CLOCK). If you can't do this experiment please discuss.

Please detail the food intake in a better way (how much they ate in the night phase, how much they ate in the light phase, adjust the food intake to the weight of the animal ...) and discuss, this is a very important point of the study. I suggest making a table with date of food intake.

Please take better advantage of the H&E, measure the number of membrane nuclei and central nuclei, remembering that eccentric exercise generates high muscle damage.

Since you have individual corticosterone values (changes) and you believe that it is responsible for your cellular results, please make correlations between changes in corticosterone and changes in protein signaling, CSA and strength level.

6. PLOS authors have the option to publish the peer review history of their article (what does this mean?). If published, this will include your full peer review and any attached files.

Reviewer #1: No

Reviewer #2: No

---

## [Author Response · Author response to Decision Letter 0]

4 Oct 2021

PONE-D-21-07409

Circadian rhythms modulate the effect of eccentric exercise on rat skeletal muscles via the mTOR signaling pathway

PLOS ONE

We wish to express our appreciation to the reviewers for their insightful comments on our manuscript. Their comments have helped us significantly improve our paper. We have attempted to address all their concerns. Our point-by-point responses to the comments are given below.

Responses to Reviewer #1

Comment: This study aimed to investigate the effect of circadian rhythms on exercise-mediated muscle anabolism. The authors concluded that the degree of eccentric exercise-mediated muscle hypertrophy in soleus muscle is time-of-day dependent and circadian rhythms of corticosterone and mTOR/p70S6K signaling affect this adaptation. Although this is an interesting project that will bring important insight to our field, there are many concerns.

Comment 1: L46-48: The authors do not follow the current dogma that mTORC1/p70S6K is not a major signaling pathway in the regulation of contraction-induced protein synthesis. Also, ref. 10 does not report the correlation between signaling and muscle hypertrophy. The authors should revise this point and further add the reference that ERK signaling contributes to the contraction-induced protein synthesis.

Response 1: Thank you for your suggestion. There are many signaling pathways that promote muscle growth, such as Insulin/IGF1-AKT-mTOR, TGFβ/myostatin/activin/BMP, β-adrenergic signaling, FGF/zinc ions, and desmosomes/Zinc ions (Sartori et al., 2021). However, until recently, the mTORC1/p70S6K (Kotani et al., 2021; Ashida et al., 2018; Ribeiro et al., 2017) and ERK (Williamson, et al., 2003; Takegaki et al., 2019) signaling have still been used as an indicator for the synthesis of muscle proteins. Therefore, we believe that both mTORC1/p70S6K and ERK signaling are important. We will investigate the other contraction-induced protein synthesis pathways in the future.

We have included the abovementioned information in the revised text and cited the relevant references (p. 3, lines 47–48; p. 16–17, lines 273–280).

References:

Sartori, R., Romanello, V., & Sandri, M. (2021). Mechanisms of muscle atrophy and hypertrophy: Implications in health and disease. Nature Communications, 12(1), 1-12.

Kotani, T., Takegaki, J., Tamura, Y., Kouzaki, K., Nakazato, K., & Ishii, N. (2021). The effect of repeated bouts of electrical stimulation-induced muscle contractions on proteolytic signaling in rat skeletal muscle. Physiological Reports, 9(9), e14842.

Ashida, Y., Himori, K., Tatebayashi, D., Yamada, R., Ogasawara, R., & Yamada, T. (2018). Effects of contraction mode and stimulation frequency on electrical stimulation-induced skeletal muscle hypertrophy. Journal of Applied Physiology, 124(2), 341-348.

Ribeiro, M. B. T., Guzzoni, V., Hord, J. M., Lopes, G. N., de Cássia Marqueti, R., de Andrade, R. V., ... & Durigan, J. L. Q. (2017). Resistance training regulates gene expression of molecules associated with intramyocellular lipids, glucose signaling and fiber size in old rats. Scientific Reports, 7(1), 1-13.

Williamson, D., Gallagher, P., Harber, M., Hollon, C., & Trappe, S. (2003). Mitogen-activated protein kinase (MAPK) pathway activation: effects of age and acute exercise on human skeletal muscle. The Journal of Physiology, 547(3), 977-987.

Takegaki, J., Sase, K., & Fujita, S. (2019). Repeated bouts of resistance exercise attenuate mitogen-activated protein-kinase signal responses in rat skeletal muscle. Biochemical and Biophysical Research Communications, 520(1), 73-78.

Comment 2: L68: Why did the authors only investigate soleus muscle? This point significantly reduces the value of this paper. Generally, downhill running affects not only soleus muscle but also gastrocnemius (and plantalis) muscle and therefore the authors should add the data of gastrocnemius at least the results of chronic training.

Response 2: Thank you for your suggestion. We have presented the observations made in the gastrocnemius and plantalis muscles in Table 1 (p. 11–12, lines 192–196). We have indicated that only soleus, for which interaction was found, was used in the analysis because there was no interaction or other effect of downhill training on gastrocnemius or plantalis.

Comment 3: L84-87: Was exercise intensity constant?

Response 3: Yes, it was constant. The program for each training session was 16 m/min (incline: -16°) and 90 minutes of exercise time.

Comment 4: L95-96: Ref. 4 (your previous study) reported that mTOR signaling is high at ZT6 as compared with at ZT18 in plantaris but not soleus.

Response 4: In the SOL, there was no significant difference, but there was a 49.5% difference observed between the two points near the highest and lowest values; thus, even if there was no difference at rest (or even if the difference was not large), the effects of exercise could still be different.

Comment 5: L100: Why did the authors use different exercise protocols between acute and chronic exercise?

Response 5: Although the increase and duration of the signal may be different when the time is shortened, the time is shortened because it helps obtain an understanding of the signaling pathways involved. We decided that it is not necessary to use the same protocol in order to see the response reflected one time because each response will be different during the adaptation process.

Comment 6: L123-: Is 99% glycerol correct? What does “samples containing total protein” mean?

Response 6: Thank you for pointing this out. The final concentration was 30% glycerol, 5% 2-mercaptoethanol, 2.3% SDS, 62.5 mm Tris–HCl pH 6.8, and bromphenol blue. We have corrected the relevant text in the revised manuscript (p. 8, lines 128–129).

“Samples containing total protein” are samples of mixtures of protein extracts and sample buffer.

Comment 7: L156-162: Why is there no difference in body weight between the control and exercise training groups when the control rats eat a lot and do not exercise? Need discussion and add fat mass data if possible.

Response 7: We agree with the reviewer’s contention, and acknowledge that the suggestion made by the reviewer is valuable. We have added the fat mass data (revised Fig. 1) and have discussed this issue in the Discussion section of the revised manuscript (p. 15, lines 248–260).

We noticed that the information on food intake in the original manuscript was incorrect. We apologize for this oversight and have corrected it in the revised manuscript. The food intake in the control group was almost the same as that in the training group; however, rats in the control group in this study had more body fat (epididymal white adipose tissue) than those in the training group.

Comment 8: L187-191: mTOR S2448 phosphorylation does not necessarily associate mTORC1 activity. The authors can evaluate mTORC1 activity more appropriately to measure both p70S6K and 4E-BP1 phosphorylation.

Response 8: We agree with the valuable comment made by the reviewer. We added 4E-BP1 data and modified the relevant text in the revised manuscript (revised Fig. 3 and p. 17, lines 281–290).

Comment 9: L202-204: The authors report only a %change in serum corticosterone concentration. Please add absolute values at pre- and post-exercise.

Response 9: Thank you for your suggestion. We have added the serum corticosterone concentration data (revised Fig. 4 and p. 14, lines 225–239).

Comment 10: L217-226: Resistance exercise increases mTORC1 signaling for more than 24h after exercise while endurance exercise activates mTORC1 signaling only immediately after exercise. The results of this study are in accordance with the signaling responses of endurance exercise rather than resistance exercise, suggesting that factors other than mTORC1 signaling associate downhill running-induced muscle hypertrophy.

Response 10: Thank you for your keen analysis. The signaling pattern described in the comment is certainly true for humans, but that in animals can also be degraded in a relatively short time in response to hypertrophy-inducing movements, such as electrical stimulation (Bolster et al., 2003). However, we believe that the timing of the exercise is also important. In this study, we assessed the effects of a bout of exercise on Wistar rats to check whether exercising at different times (ZT6 or ZT18) had different effects on mTOR/p70S6K signaling.

References:

Bolster, D. R., Kubica, N., Crozier, S. J., Williamson, D. L., Farrell, P. A., Kimball, S. R., & Jefferson, L. S. (2003). Immediate response of mammalian target of rapamycin (mTOR)‐mediated signalling following acute resistance exercise in rat skeletal muscle. The Journal of physiology, 553(1), 213-220.

Comment 11: L228: Ref. 4 does not report the role of ERK signaling in the mechanisms of muscle hypertrophy.

Response 11: Ref. 4 was not cited on line 228. In Ref. 4, we described that the physiological significance of circadian variation in ERK phosphorylation in the cardiac and plantaris muscles remains unclear. Based on the results of transient exercise at different times in this study, there was no difference in ERK signaling, suggesting that this signaling is less likely as a pathway for exercise-induced muscle hypertrophy in the light period. Nevertheless, the results of ERK signaling in this study can prove that the intensity of exercise performed at different times is the same.

We have modified the relevant text in the revised manuscript because our explanation may be inadequate (p. 16–17, lines 277–280).

Comment 12: L236-253: Regarding corticosterone, the authors state that serum corticosterone may be associated mTOR/p70S6K signaling, as it has a role in muscle protein breakdown. Please discuss more specifically with the references. How muscle protein breakdown affects mTOR signaling?

Serum corticosterone response may affect the mTOR signaling response but how about the resting condition? Not resting (absolute value) but only responses affect mTOR signaling?

It is interesting and important to measure REDD1 to know the mechanisms of reduced mTORC1 responses.

Response 12:

Thank you for your suggestion. Considering muscle protein degradation affects mTOR signaling, we have added a discussion to the revised manuscript (p. 17–18, lines 297–306).

The serum corticosterone concentration in the light period (ZT6) group was generally higher than that in the dark period (ZT18) group, possibly due to the interconnection between circadian rhythm and corticosterone concentration that peaked around ZT11-13. In contrast, the corticosterone peak in the dark period (ZT18) group was past and the concentration had lowered. The serum corticosterone levels in LTR and DLR groups immediately after exercise were not significantly different, but the rate of increase in corticosterone concentration immediately after exercise was greater in the dark-phase group than in the light-phase group (revised Fig. 4). These observations tentatively suggest that corticosterone has a slightly antagonistic impact on mTOR/p70S6K signaling during the light phase, explaining the increase in exercise-induced p70S6K phosphorylation and subsequent muscle hypertrophy during that time of day (p. 18–19, lines 310–317). However, the underlying mechanisms are unknown, and would be investigated in the future.

We tried to analyze the expression of REDD1 (35 kDa) using antibodies against it (10638-1-AP) purchased from Proteintech Group, Inc., but could not detect clear bands (as shown in the figure below). 

Comment 13: L260-265: Obviously, exercise type and fiber type affect the signaling responses. However, the authors suggested that different adaptation to the training is led by serum corticosterone (systemic factor). Therefore, the authors should discuss not only signaling but also systemic factors.

Response 13:

Thank you for your suggestion. We have added the following discussion to the revised manuscript (p. 19, lines 334–338):

Moreover, hormones as indicators of systemic factors also affect muscle mass; for example, differences in hormone levels in the body (testosterone:corticosterone ratio) or differences in hormone levels caused by exercise. Therefore, the effect of systemic factors on muscle mass cannot be ignored (Crowley & Matt, 1996; Guzzoni, 2019).

Comment 14: L268: Which paper shows that the timing of feeding is important to induce muscle hypertrophy?

Response 14: Thank you for this query. We have cited the following reference (p. 20, line 341):

Phillips, S. M., Tipton, K. D., Aarsland, A., Wolf, S. E., & Wolfe, R. R. (1997). Mixed muscle protein synthesis and breakdown after resistance exercise in humans. American Journal of Physiology, 273(1), E99-E107.

Comment 15: L269: Ref. 30 is a review paper. So please add the original article(s).

Response 15: Thank you for your suggestion. We have cited the following original article (p. 20, line 344):

Burd, N. A., West, D. W., Moore, D. R., Atherton, P. J., Staples, A. W., Prior, T., ... & Phillips, S. M. (2011). Enhanced amino acid sensitivity of myofibrillar protein synthesis persists for up to 24 h after resistance exercise in young men. The Journal of Nutrition, 141(4), 568-573.

 

Responses to Reviewer #2

Comment 1: Please if it is possible to show the circadian clock proteins or the gene expression, since exercise could have modified your circadian clock (i.e Bmal1 / CLOCK). If you can't do this experiment please discuss.

Response 1: Thank you for your suggestion. Unfortunately, we could not measure the gene expression in this study because of limitation of the samples. Therefore, we have added relevant discussion about it in the revised manuscript. (p. 21, lines 367–373).

The clock gene has been reported to be involved in the structure and function of skeletal muscle (Andrews et al., 2010). It has also been demonstrated that exercise affects the circadian clock (Bmal1/CLOCK) of skeletal muscle (Wolff et al., 2012). Based on these studies, it may be possible to influence muscles from exercise via the clock genes. As per recent studies, some pathways are likely to influence Per1 by glucocorticoids via REDD1 (Saracino et al., 2019). Another pathway possibly affects BMAL1 through the activation of mTORC1 (Dadon-Freiberg et al., 2021). However, further research is needed to confirm whether these pathways affect the muscle mass.

Comment 2: Please detail the food intake in a better way (how much they ate in the night phase, how much they ate in the light phase, adjust the food intake to the weight of the animal ...) and discuss, this is a very important point of the study. I suggest making a table with date of food intake.

Response 2: Thank you for your comments. We measured food intake, without food and water restrictions, after a single training session of 3 days. The weekly food intake for each group is shown in the revised figure 1. Unfortunately, we are unable to present the detailed feed intake. To avoid disrupting the biological rhythms and to minimize non-training stresses (entering and exiting the animal house, turning lights on and off, etc.) in the experimental animals, we took only a limited number of measurements. It has also been previously reported that exposure to light and food restriction alters the periodicity of endocrine, body temperature, and activity in experimental animals (Mohawk et al., 2007; Wideman et al., 2009; Depres-Brummer et al., 1995; Krieger et al., 1974). In addition, it is reported that factors like exercise and food restriction were stressful for experimental animals and affected the secretion of corticosterone (Fediuc, 2006; Heiderstadt, 2000). However, although we do not know exactly when the animals ate, there was no notable difference observed in food intake.

Comment 3: Please take better advantage of the H&E, measure the number of membrane nuclei and central nuclei, remembering that eccentric exercise generates high muscle damage.

Response 3: We are thankful for the valuable suggestion. We have added the data on the central nucleus as an indicator of regenerating muscle (revised Fig. 2) and have discussed this issue in the Discussion section of the revised manuscript (p. 21, lines 358–366).

Upon the onset of muscle damage, satellite cells are activated; they proliferate and differentiate, eventually fusing with other satellite cells to form myofibers with central nuclei (Sasaki et al., 2007; Yu et al., 2017). Therefore, the central nuclei (shown as the ratio of the centronucleated fiber) are used as an indicator of regenerating muscle. It was previously reported that the central nucleus of the soleus muscle of rats was clearly observed after 5 days of a bout of downhill running (Yu et al., 2017). However, in this study, there was no difference in the central nucleus in any of the groups because we used the same training intensity and training time for 8 weeks. Therefore, the muscles would be accustomed to the training stimulus, and damage would be less likely to occur.

Comment 4: Since you have individual corticosterone values (changes) and you believe that it is responsible for your cellular results, please make correlations between changes in corticosterone and changes in protein signaling, CSA and strength level.

Response 4: Thank you for your suggestion. We acknowledge that the analyses suggested by you would provide great insights. However, we could not determine the correlations between the variables because these data were obtained from different rats (i.e., they are not matching data).

---

## [Decision Letter · Decision Letter 1]

6 Dec 2021

PONE-D-21-07409R1Circadian rhythms modulate the effect of eccentric exercise on rat skeletal muscles via the mTOR signaling pathwayPLOS ONE

Dear Dr. Naito,

Thank you for submitting your manuscript to PLOS ONE. After careful consideration, we feel that it has merit but does not fully meet PLOS ONE’s publication criteria as it currently stands. Therefore, we invite you to submit a revised version of the manuscript that addresses the points raised during the review process by the reviewer #1.

We look forward to receiving your revised manuscript.

Kind regards,

Atsushi Asakura, Ph.D

Academic Editor

PLOS ONE

Reviewers' comments:

Reviewer's Responses to Questions

**Comments to the Author**

1. If the authors have adequately addressed your comments raised in a previous round of review and you feel that this manuscript is now acceptable for publication, you may indicate that here to bypass the “Comments to the Author” section, enter your conflict of interest statement in the “Confidential to Editor” section, and submit your "Accept" recommendation.

Reviewer #1: (No Response)

Reviewer #2: All comments have been addressed

2. Is the manuscript technically sound, and do the data support the conclusions?

Reviewer #1: Yes

Reviewer #2: Yes

3. Has the statistical analysis been performed appropriately and rigorously? 

Reviewer #1: (No Response)

Reviewer #2: Yes

4. Have the authors made all data underlying the findings in their manuscript fully available?

Reviewer #1: Yes

Reviewer #2: Yes

5. Is the manuscript presented in an intelligible fashion and written in standard English?

Reviewer #1: Yes

Reviewer #2: Yes

6. Review Comments to the Author

Reviewer #1: Comment 1: L46-48: The authors do not follow the current dogma that mTORC1/p70S6K is not a major signaling pathway in the regulation of contraction-induced protein synthesis. Also, ref. 10 does not report the correlation between signaling and muscle hypertrophy. The authors should revise this point and further add the reference that ERK signaling contributes to the contraction-induced protein synthesis.

Response 1: Thank you for your suggestion. There are many signaling pathways that promote muscle growth, such as Insulin/IGF1-AKT-mTOR, TGFβ/myostatin/activin/BMP, β-adrenergic signaling, FGF/zinc ions, and desmosomes/Zinc ions (Sartori et al., 2021). However, until recently, the mTORC1/p70S6K (Kotani et al., 2021; Ashida et al., 2018; Ribeiro et al., 2017) and ERK (Williamson, et al., 2003; Takegaki et al., 2019) signaling have still been used as an indicator for the synthesis of muscle proteins. Therefore, we believe that both mTORC1/p70S6K and ERK signaling are important. We will investigate the other contraction-induced protein synthesis pathways in the future.

We have included the abovementioned information in the revised text and cited the relevant references (p. 3, lines 47–48; p. 16–17, lines 273–280).

It’s not scientific. They might use mTORC1 and ERK as an indicator but mTORC1/p70S6K is not a major signaling pathway in the regulation of contraction-induced protein synthesis (PMID: 26227152, 30509128. Please note that PMID: 30509128 reported that while the contraction-induced protein synthesis is mTORC1-independent, muscle hypertrophy is mTORC1-dependent).

Also, there is no evidence that ERK regulates contraction-induced muscle protein synthesis, and rather that is denied (PMID: 23077579).

Therefore, of course, the authors can use mTORC1 as an indicator but state and discuss carefully based on the fact because the authors did not measure muscle protein synthesis.

Comment 2: L68: Why did the authors only investigate soleus muscle? This point significantly reduces the value of this paper. Generally, downhill running affects not only soleus muscle but also gastrocnemius (and plantalis) muscle and therefore the authors should add the data of gastrocnemius at least the results of chronic training.

Response 2: Thank you for your suggestion. We have presented the observations made in the gastrocnemius and plantalis muscles in Table 1 (p. 11–12, lines 192–196). We have indicated that only soleus, for which interaction was found, was used in the analysis because there was no interaction or other effect of downhill training on gastrocnemius or plantalis.

Please discuss why exercise in this study induced muscle hypertrophy only in soleus muscle despite eccentric exercise predominantly activates fast-twitch fibers. Based on this result, the author should change the title from “rat skeletal muscle” to “rat soleus muscle”

Comment 4: L95-96: Ref. 4 (your previous study) reported that mTOR signaling is high at ZT6 as compared with at ZT18 in plantaris but not soleus.

Response 4: In the SOL, there was no significant difference, but there was a 49.5% difference observed between the two points near the highest and lowest values; thus, even if there was no difference at rest (or even if the difference was not large), the effects of exercise could still be different.

Even if there was a relatively large difference, if there was no significant difference, the authors should not be stated that there is a difference.

Comment 10: L217-226: Resistance exercise increases mTORC1 signaling for more than 24h after exercise while endurance exercise activates mTORC1 signaling only immediately after exercise. The results of this study are in accordance with the signaling responses of endurance exercise rather than resistance exercise, suggesting that factors other than mTORC1 signaling associate downhill running-induced muscle hypertrophy.

Response 10: Thank you for your keen analysis. The signaling pattern described in the comment is certainly true for humans, but that in animals can also be degraded in a relatively short time in response to hypertrophy-inducing movements, such as electrical stimulation (Bolster et al., 2003). However, we believe that the timing of the exercise is also important. In this study, we assessed the effects of a bout of exercise on Wistar rats to check whether exercising at different times (ZT6 or ZT18) had different effects on mTOR/p70S6K signaling.

Bolster et al 2003 used a squat model. They used ES to impose a jump but not to directly stimulate muscle activation. This squat model does not induce muscle hypertrophy and is currently not used as a resistance training model.

Comment 12: L236-253: Regarding corticosterone, the authors state that serum corticosterone may be associated mTOR/p70S6K signaling, as it has a role in muscle protein breakdown. Please discuss more specifically with the references. How muscle protein breakdown affects mTOR signaling?

Serum corticosterone response may affect the mTOR signaling response but how about the resting condition? Not resting (absolute value) but only responses affect mTOR signaling?

It is interesting and important to measure REDD1 to know the mechanisms of reduced mTORC1 responses.

Response 12:

Thank you for your suggestion. Considering muscle protein degradation affects mTOR signaling, we have added a discussion to the revised manuscript (p. 17–18, lines 297–306).

The main point was that how muscle protein breakdown affects mTOR. It is well known that corticosterone regulates protein breakdown and mTOR, respectively, but it is unclear how protein breakdown affects mTOR. Please delete the following sentence: “as it has a well-known role in the breakdown of muscle proteins.

Comment 14: L268: Which paper shows that the timing of feeding is important to induce muscle hypertrophy?

Response 14: Thank you for this query. We have cited the following reference (p. 20, line 341):

Phillips, S. M., Tipton, K. D., Aarsland, A., Wolf, S. E., & Wolfe, R. R. (1997). Mixed muscle protein synthesis and breakdown after resistance exercise in humans. American Journal of Physiology, 273(1), E99-E107.

They did not investigate the timing of feeding. Also, they did not measure muscle size.

Finally, please consider changing the title from “via the mTOR…” to “possibly via the mTOR…” or delete “via the mTOR…” because the authors have not proven causation.

Reviewer #2: (No Response)

7. PLOS authors have the option to publish the peer review history of their article (what does this mean?). If published, this will include your full peer review and any attached files.

Reviewer #1: No

Reviewer #2: No

---

## [Author Response · Author response to Decision Letter 1]

15 Jan 2022

We appreciate the insightful comments from the reviewers. Their comments have helped us improve our paper. We have attempted to address all their concerns. Our point-by-point responses to the comments are given below.

Reviewer #1: 

Comment 1: L46-48: The authors do not follow the current dogma that mTORC1/p70S6K is not a major signaling pathway in the regulation of contraction-induced protein synthesis. Also, ref. 10 does not report the correlation between signaling and muscle hypertrophy. The authors should revise this point and further add the reference that ERK signaling contributes to the contraction-induced protein synthesis.

Response 1: Thank you for your suggestion. There are many signaling pathways that promote muscle growth, such as Insulin/IGF1-AKT-mTOR, TGFβ/myostatin/activin/BMP, β-adrenergic signaling, FGF/zinc ions, and desmosomes/Zinc ions (Sartori et al., 2021). However, until recently, the mTORC1/p70S6K (Kotani et al., 2021; Ashida et al., 2018; Ribeiro et al., 2017) and ERK (Williamson, et al., 2003; Takegaki et al., 2019) signaling have still been used as an indicator for the synthesis of muscle proteins. Therefore, we believe that both mTORC1/p70S6K and ERK signaling are important. We will investigate the other contraction-induced protein synthesis pathways in the future.

We have included the abovementioned information in the revised text and cited the relevant references (p. 3, lines 47–48; p. 16–17, lines 273–280).

Comment 1-1: It’s not scientific. They might use mTORC1 and ERK as an indicator but mTORC1/p70S6K is not a major signaling pathway in the regulation of contraction-induced protein synthesis (PMID: 26227152, 30509128. Please note that PMID: 30509128 reported that while the contraction-induced protein synthesis is mTORC1-independent, muscle hypertrophy is mTORC1-dependent). Also, there is no evidence that ERK regulates contraction-induced muscle protein synthesis, and rather that is denied (PMID: 23077579). Therefore, of course, the authors can use mTORC1 as an indicator but state and discuss carefully based on the fact because the authors did not measure muscle protein synthesis.

Response 1-1:

We thank the reviewer for this suggestion. We acknowledge that in the study by You et al. (2019), protein synthesis by mechanical loading is mTORC1-independent, but muscle hypertrophy is mTORC1-dependent. However, some previous studies have shown that exercise-induced protein synthesis involves mTORC1. A study in mice by Ogasawara et al. (2018) reported that resistance exercise (transcutaneous electrical stimulation to induce muscle contraction) increases muscle protein synthesis via rapamycin-sensitive mTORC1 and rapamycin-insensitive mTORC1 or mTORC2. The difference in these results might be due to the differences in the pathways of muscle protein synthesis induced by different stimuli leading to muscle hypertrophy like electrical stimulation or myotenectomy. In this study, we did not measure muscle protein synthesis; therefore, the relationship between mTORC1 and muscle protein synthesis by downhill running is currently unknown. Following the reviewer's advice, we have described this in the Discussion section.

We have added the following discussion to the revised manuscript (p. 19–20, lines 327–339):

“Previous studies have shown a relationship between mTOR/p70S6K and muscle protein synthesis. A mouse study by Ogasawara et al. (2018) reported that the promotion of muscle protein synthesis by resistance exercise (transcutaneous electrical stimulation to induce muscle contraction) is regulated by mTOR (both rapamycin-sensitive mTORC1 and rapamycin-insensitive mTORC1 or mTORC2). However, a recent study reported that exercise-induced protein synthesis is independent of mTOR/p70S6K. You et al. (2019) reported that protein synthesis by mechanical loading is mTORC1-independent in the case of myotenectomy-induced muscle hypertrophy in mice. Different experimental methods of inducing muscle hypertrophy-like electrical stimulation or myotenectomy-induced muscle hypertrophy may also result in different pathways of muscle protein synthesis. Therefore, the relationship between mTOR/p70S6K and muscle protein synthesis remains unclear. In this study also we have not elucidated the relationship between mTOR/p70S6K as we have not measured muscle protein synthesis.”

References:

Ogasawara, R., & Suginohara, T. (2018). Rapamycin‐insensitive mechanistic target of rapamycin regulates basal and resistance exercise‐induced muscle protein synthesis. The FASEB Journal, 32(11), 5824-5834.

You, J. S., McNally, R. M., Jacobs, B. L., Privett, R. E., Gundermann, D. M., Lin, K. H., ... & Hornberger, T. A. (2019). The role of raptor in the mechanical load‐induced regulation of mTOR signaling, protein synthesis, and skeletal muscle hypertrophy. The FASEB Journal, 33(3), 4021-4034.

Comment 2: L68: Why did the authors only investigate soleus muscle? This point significantly reduces the value of this paper. Generally, downhill running affects not only soleus muscle but also gastrocnemius (and plantalis) muscle and therefore the authors should add the data of gastrocnemius at least the results of chronic training.

Response 2: Thank you for your suggestion. We have presented the observations made in the gastrocnemius and plantalis muscles in Table 1 (p. 11–12, lines 192–196). We have indicated that only soleus, for which interaction was found, was used in the analysis because there was no interaction or other effect of downhill training on gastrocnemius or plantalis.

Comment 2-1: Please discuss why exercise in this study induced muscle hypertrophy only in soleus muscle despite eccentric exercise predominantly activates fast-twitch fibers. Based on this result, the author should change the title from “rat skeletal muscle” to “rat soleus muscle”

Response 2-1:

We thank the reviewer for this suggestion. We have added the following description to the revised manuscript (p. 21–22, lines 368–377) and changed the title to ‘rat soleus muscle’.

The previous study on rats has shown that most of the fiber components of the soleus muscle are mobilized during downhill running (incline: -16°) (Smith et al., 1997). In addition, when low-intensity exercise (10-20 m/min) is performed with downhill running (treadmill grade 15% decline), fast-twitch muscles might be involved to a much lesser extent in the motor unit recruitment (Dudley et al., 1982). Therefore, the cause of the induced muscle hypertrophy only in the soleus muscle could be determined as only the low-intensity exercise was involved in this study. It is thought that low-intensity exercise may not reflect the effects of circadian rhythm on fast-twitch muscles as the fast-twitch muscles are scarcely stimulated by low-intensity exercise. In the future, employment of high-intensity exercise will be required to prove this.”

References:

Smith, H. K., Plyley, M. J., Rodgers, C. D., & McKee, N. H. (1997). Skeletal muscle damage in the rat hindlimb following single or repeated daily bouts of downhill exercise. International journal of sports medicine, 18(02), 94-100.

Dudley, G. A., Abraham, W. M., & Terjung, R. L. (1982). Influence of exercise intensity and duration on biochemical adaptations in skeletal muscle. Journal of applied physiology, 53(4), 844-850.

Comment 4: L95-96: Ref. 4 (your previous study) reported that mTOR signaling is high at ZT6 as compared with at ZT18 in plantaris but not soleus.

Response 4: In the SOL, there was no significant difference, but there was a 49.5% difference observed between the two points near the highest and lowest values; thus, even if there was no difference at rest (or even if the difference was not large), the effects of exercise could still be different.

Comment 4-1: Even if there was a relatively large difference, if there was no significant difference, the authors should not be stated that there is a difference.

Response 4-1:

We apologize for the inadequate explanation in our previous reply. The statistically significant difference described in our previous study (Reference no. 4) was calculated using a statistical method of the modified cosinor analysis to determine the existence of a circadian rhythm. This is different from the tests of differences between means that are usually used (e.g., t-test). Therefore, we think that the statement in our previous publication, "Circadian rhythms of signal transducers were observed in both cardiac (mTOR, p70S6K, and ERK) and plantaris (p70S6K and ERK) muscles (p < 0.05), but not in the soleus muscle." is reasonable. In addition, in the cited study (Reference no. 4), we do not claim that the mTOR signaling is high at ZT6 as compared with at ZT18 in plantaris but not soleus.

Comment 10: L217-226: Resistance exercise increases mTORC1 signaling for more than 24h after exercise while endurance exercise activates mTORC1 signaling only immediately after exercise. The results of this study are in accordance with the signaling responses of endurance exercise rather than resistance exercise, suggesting that factors other than mTORC1 signaling associate downhill running-induced muscle hypertrophy.

Response 10: Thank you for your keen analysis. The signaling pattern described in the comment is certainly true for humans, but that in animals can also be degraded in a relatively short time in response to hypertrophy-inducing movements, such as electrical stimulation (Bolster et al., 2003). However, we believe that the timing of the exercise is also important. In this study, we assessed the effects of a bout of exercise on Wistar rats to check whether exercising at different times (ZT6 or ZT18) had different effects on mTOR/p70S6K signaling.

Comment 10-1: Bolster et al 2003 used a squat model. They used ES to impose a jump but not to directly stimulate muscle activation. This squat model does not induce muscle hypertrophy and is currently not used as a resistance training model.

Response 10-1:

We apologize for the lack of explanation. As pointed out by the reviewer, the study by Bolster et al. (2003) is an experiment in which rats were made to jump using electrical stimulation. In addition, a previous study by Nader et al. (2001) used low-frequency electrical stimulation to increase mTORC1 signaling (p70S6K), which returned to baseline 6 hours after stimulation. We believe that different exercise intensities cause distinct changes in signaling.

Reference:

Nader, G. A., & Esser, K. A. (2001). Intracellular signaling specificity in skeletal muscle in response to different modes of exercise. Journal of applied physiology, 90(5), 1936-1942.

Comment 12: L236-253: Regarding corticosterone, the authors state that serum corticosterone may be associated mTOR/p70S6K signaling, as it has a role in muscle protein breakdown. Please discuss more specifically with the references. How muscle protein breakdown affects mTOR signaling?

Serum corticosterone response may affect the mTOR signaling response but how about the resting condition? Not resting (absolute value) but only responses affect mTOR signaling?

It is interesting and important to measure REDD1 to know the mechanisms of reduced mTORC1 responses.

Response 12:

Thank you for your suggestion. Considering muscle protein degradation affects mTOR signaling, we have added a discussion to the revised manuscript (p. 17–18, lines 297–306).

Comment 12-1: The main point was that how muscle protein breakdown affects mTOR. It is well known that corticosterone regulates protein breakdown and mTOR, respectively, but it is unclear how protein breakdown affects mTOR. Please delete the following sentence: “as it has a well-known role in the breakdown of muscle proteins.

Response 12-1:

As per the reviewer’s suggestion, we have now deleted the text (p. 17, lines 291–293).

Comment 14: L268: Which paper shows that the timing of feeding is important to induce muscle hypertrophy?

Response 14: Thank you for this query. We have cited the following reference (p. 20, line 341):

Phillips, S. M., Tipton, K. D., Aarsland, A., Wolf, S. E., & Wolfe, R. R. (1997). Mixed muscle protein synthesis and breakdown after resistance exercise in humans. American Journal of Physiology, 273(1), E99-E107.

Comment 14-1: They did not investigate the timing of feeding. Also, they did not measure muscle size.

Response 14-1:

Thank you for pointing this out. We have now replaced the references. We have also revised the text to avoid misunderstanding in the revised manuscript as follows (p. 20, lines 350–352):

“Although previous studies in human subjects have demonstrated that the timing of feeding post-exercise (1-3 hours after training) plays an important role in promoting protein synthesis (Rasmussen et al., 2000),…..”

Reference:

Rasmussen, B. B., Tipton, K. D., Miller, S. L., Wolf, S. E., & Wolfe, R. R. (2000). An oral essential amino acid-carbohydrate supplement enhances muscle protein anabolism after resistance exercise. Journal of applied physiology, 88(2), 386-392.

Finally, please consider changing the title from “via the mTOR…” to “possibly via the mTOR…” or delete “via the mTOR…” because the authors have not proven causation.

Response:

As per the reviewer's suggestion, we have now changed the title to “Circadian rhythms modulate the effect of eccentric exercise on rat soleus muscles.”

Reviewer #2: (No Response)

---

## [Editor Report · Decision Letter 2]

7 Feb 2022

Circadian rhythms modulate the effect of eccentric exercise on rat soleus muscles

PONE-D-21-07409R2

Dear Dr. Naito,

We’re pleased to inform you that your manuscript has been judged scientifically suitable for publication and will be formally accepted for publication once it meets all outstanding technical requirements.

Kind regards,

Atsushi Asakura, Ph.D

Academic Editor

PLOS ONE
---

## [Editor Report · Acceptance letter]

16 Feb 2022

PONE-D-21-07409R2 

Circadian rhythms modulate the effect of eccentric exercise on rat soleus muscles 

Dear Dr. Naito:

I'm pleased to inform you that your manuscript has been deemed suitable for publication in PLOS ONE. Congratulations! Your manuscript is now with our production department. 

Kind regards, 

on behalf of

Dr. Atsushi Asakura 

Academic Editor

PLOS ONE